

**A 150-year record of phytoplankton community succession**
**controlled by hydroclimatic variability in a tropical lake**
**K. A. Yamoah[1], N. Callac[1], E. Chi Fru[1], B. Wohlfarth[1], A. Wiech[1] A.**
**Chabangborn[2] and R. H. Smittenberg[1]**
[1]Department of Geological Sciences and Bolin Centre for Climate Research, Stockholm
University, 10691 Stockholm, Sweden
[2]Departments of Geology, Faculty of Science, Chulalongkorn University, Bangkok, 10330,
Thailand
Correspondence to: K. A. Yamoah (kweku.yamoah@geo.su.se)



**Abstract**
Climate and human-induced environmental change promotes biological regime shifts between
alternate stable states, with implications for ecosystem resilience, function and services.
While this has been shown for recent microbial communities, the long-term response of
microbial communities has not been investigated in detail. This study investigated the decadal
variations in phytoplankton communities in a ~150 year long sedimentary archive of Lake
Nong Thale Prong (NTP), southern Thailand using a combination of DNA and lipid
biomarkers techniques. Reconstructed drier climate from ~1857-1916 Common Era (CE)
coincided with oligotrophic lake water conditions and dominance of the green algae
*Botryococcus braunii,* producing characteristic botryococcene lipids. A change to higher
silica (Si) input ~1916 CE, which was related to increased rainfall concurs with an abrupt
takeover by diatom blooms lasting for 50 years. Since the 1970s more eutrophic conditions
prevailed, which was likely caused by increased levels of anthropogenic phosphate (P), aided
by increased lake stratification caused by somewhat dryer conditions. The eutrophic
conditions led to increased primary productivity consisting again of a *Botryococcus sp.,*
though this time not producing the botryococcene lipids. Moreover, *Cyanobacteria* became
dominant. Our results indicate that a combined DNA and lipid biomarker approach provides
an efficient way to allow tracking centennial-scale hydroclimate and anthropogenic feedback
processes in lake ecosystems.

29



## 1 Introduction

Natural and anthropogenically induced climatic change is often cited as the main factor controlling ecosystem dynamics (Scheffer et al., 2001; Malmqvist et al., 2008; Woodward et al., 2010). The resulting environmental effects easily observed today in large plant and animal communities (Scheffer et al., 2001) have been discussed as causing regime shifts (Folke et al., 2004) and/or introducing alternative ecosystem states (Dent et al., 2002). On longer scales, climate-induced changes in microbial communities might in effect affect the capture and storage of $CO_2$ (Zeglin et al., 2013). However, predicting the effects of changing climate on shifts in microbial community structures even on short-time scales is challenging (Lotter and Birks, 1997; Woodward et al., 2010). This is partially due to the difficulty of coupling micro-scale external and internal ecosystem variables to specific microbial communities on longer time scales.

Consequently, most estimates are based on small-scale and short-term experimental data and on easily manipulated ecosystems such as soils (Landesman and Dighton, 2010; Cregger et al., 2012; Kuffner et al., 2012; Zeglin et al., 2013). Very little is therefore known about how natural microbial ecosystems respond to climatic changes over longer time scales and whether the associated feedbacks due to climate shifts promote regime shifts and alternative stable states. Moreover, interactions between external and internal ecosystem regulators still remain relatively unknown.

Because of their vital position at the base of the food chain (Carpenter et al., 1987; Young et al., 2013), phytoplankton constitutes one of the most ecologically important groups of microorganisms in aquatic ecosystems (Yin et al., 2011). The activity of these microorganisms is directly coupled to climatic changes through $CO_2$ drawdown, organic matter availability and oxygen production, factors that are necessary for the functioning of an entire ecosystem. Since phytoplankton communities are sensitive to environmental stressors (Woodward et al., 2010; Häder and Gao, 2015), natural and/or human-induced disturbances may result in far-reaching consequences for the nutrient status of lakes, where phytoplankton is the primary producer (Yin et al., 2011). For instance, the demise of the botryococcene lipid-producing green algae *Botryococcus braunii* was linked to early eutrophication in a Norwegian fjord (Smittenberg *et al.,* 2005).

Other studies have related variations in phytoplankton community structure, abundance and function to changes in lake trophic status (Ravasi et al., 2012; Hou et al., 2014).





Consequently, the emerging paradigm suggests that the structuring of phytoplankton
communities, characterized by potential successional shifts in population dynamics, may
serve as a tracer of the trophic status in lakes (Mackay et al., 2003; De Senerpont Domis et al.,
2007). Yet, changes in phytoplankton communities and productivity are to a large extent also
influenced by complex internal and external controls (Kamenir et al., 2008; Wang et al.,
2015), which however still remain to be elucidated.
Although there is a growing understanding of the factors that influence microbial
communities in lake ecosystems (Thyssen et al., 2011; Wang et al., 2015), the broader
linkages between different microbial groups and their response to past environmental
conditions are poorly understood. This is partly due to the lack of suitable proxies that can
capture and distinguish between the diverse parameters impacting microbial ecosystem
structure.
Lipid biomarkers specific for various types of microbes provide an important proxy of
microbial ecosystem structure and have therefore been employed in the reconstruction of past
ecosystems preserved in sedimentary records (Zimmerman and Canuel, 2000; Coolen et al.,
2004; Smittenberg et al., 2005). Lipids are an integral part of the cell membranes of both
prokaryotes and eukaryotes. They aid as structural support and as storage compounds within
various microbial cells (Jungblut et al., 2009), and can be specific for certain groups or
species. Moreover, lipid biomarkers can incorporate information of the chemical environment
in which they have formed via their relative abundance or isotopic composition. Because
lipids are resistant to postmortem biodegradation, ecological variations through time can be
reconstructed (Zimmerman and Canuel, 2000; Coolen et al., 2004; Smittenberg et al., 2005)
by tracking their occurrence and abundance over longer time scales. Moreover, when coupled
to hydroclimate variables such as hydrogen isotopes of leaf waxes ($\delta D_{wax}$), a proxy for
precipitation (e.g. Niedermeyer et al., 2014), microbial lipids may help underpin the impact of
past climates (wetter/drier) on microbial ecosystem changes (see Supplement).
Quantitative polymerase chain reactions (qPCR) of specific DNA of living organisms and
well-preserved DNA in lake sediments are excellent tools to assess present and past microbial
ecosystem structures (Coolen and Gibson, 2009; Ravasi et al., 2012; Hou et al., 2014). These
analyses provide specific proof of recent and past biological processes by targeting specific
microbial taxa and key genes involved in various metabolic pathways (Takai and Horikoshi,
2000; Hou et al., 2014).





Here we explore the novel combination of biolipid analysis, $\delta D_{wax}$, qPCR, bulk isotopes of C
and N, and sedimentary geochemistry to reconstruct phytoplankton community dynamics
over a 150-year history of Lake Nong Thale Prong (NTP), Southern Thailand. Together, these
proxies allow unraveling how external forcing (hydroclimate and human impact) influences
internal abiotic feedback processes, which in turn control phytoplankton regime shifts.
**2    Materials and Methods**
**2.1    Study area, fieldwork, sediment sampling and dating**
NTP (17˚ 11`N, 99˚ 23`E) is a shallow (<7 m water depth), small (~210 m$^2$) sinkhole lake
located on the Thai-Malay Penninsula in southern Thailand (Fig. 1) at ~60 m above sea level
(Snansieng et al., 1976) (see Supplement). Prior to coring, a preliminary assessment of NTP
was conducted based on the catchment geology and topography, basin size and water depth.
Sounding in different parts of the lake showed that the deepest part was in the northeast and
that there was little variability in the distribution of sedimentary materials throughout the lake.
The two sediment cores (~74 cm length) were retrieved from the deepest part of the lake in
January 2012 using an HTH gravity corer (70 mm diameter, 1 m length). The sediment
presented a strong sulphidic smell, suggesting anoxic conditions. Visually, the cores were
lithologically similar. Therefore further analysis was performed on one core, while the other
core was archived. The sediment core was sliced onsite into 1 cm sub-samples, packed in
sterile plastics bags and chilled with ice. Such temperatures have been shown to be low
enough to inactivate and preserve whole tropical communities in sediments without the need
for dramatic freezing because of a narrower temperature range of activity (Robador et al.,
2015).  After arrival at the Department of Geological Sciences, Stockholm University the
samples were immediately frozen (-18˚C) until further analysis. The sampled sediment
sequence was dated using $^{210}$Pb (46.51 keV) and $^{226}$Rn (295.2 keV) on an EG&G ORTEC®
co-axial low energy photo spectrometer (LEPS) fitted with a high-purity germanium crystal
(see Supplement).
**2.2    Bulk biogeochemical analysis**
A total of 15 sub-samples were taken at different core depths for a low-resolution
quantification of total organic carbon (TOC), total nitrogen (TN), and carbon and nitrogen
isotopes ($\delta^{13}C_{bulk\ org}$ and $\delta^{15}N_{bulk\ org}$). Samples were freeze-dried and homogenized before





analysis. For the stable isotope measurements, the samples were pre-treated with HCl to
remove carbonate carbon before analysis on a Carlo Erba NC2500 elemental analyzer,
coupled to a Finnigan MAT Delta+ mass spectrometer. $\delta^{13}C_{bulk\ org}$ and $\delta^{15}N_{bulk\ org}$ values are
reported in parts per mille (‰) relative to the Vienna PeeDee Belemnite (VPDB, for C) and
standard air (for N), respectively, with an analytical error of ±0.15‰.
Relative estimates of the chemical composition of the sediments were obtained by elemental
mapping using Environmental Scanning Electron Microscopy (FEI, Quanta FEG 650)-
Electron Dispersive Spectroscopy (ESEM-EDS). An aliquot of the dried sediment was
mounted on aluminium stubs with carbon tape and imaged at 10 kV in low vacuum.
Elemental analysis was conducted in low vacuum with EDS at 30 kV. Approximately 75
elemental maps distributed over 15 samples across the entire core were acquired with the
AZtech software, at a horizontal field width of 2 mm, 512 pixels and an average frame count
of 5 with 100 μs pixel dwell time. The relative elemental abundance acquired was normalized
to 100%.

## 2.3 Biomarker analysis

After freeze-drying, powdered samples were extracted three times with a mixture of
dichloromethane and methanol (DCM–MeOH, 9:1, v/v) to obtain a combined total lipid
extract (TLE), using a microwave system (MILESTONE Ultra Wave Single Chamber
Microwave Digestion System) fitted with a LABTECH smart H150-1000 Water Chiller. The
TLE from the sediments was dried in a vacuum concentrator (Scanvac MaxiVac Beta,
Labogene ApS, Denmark) before being re-dissolved in DCM and then adsorbed onto a small
amount of silica gel. This was evaporated on a warm plate, under a very gentle stream of
nitrogen gas, and placed on top of 15 g silica gel (deactivated with 5% (wt.) $H_2O$) in 6-mL
glass SPE tubes. Hydrocarbon (FI), ketone (F2) and polar (F3) fractions were recovered with
pure hexane, a hexane and DCM mixture (1:1) and DCM–MeOH (1:1), respectively. F2 and
F3 samples were stored in the freezer for later use. The F1 fraction was analyzed on a
Shimadzu GCMS-QP2010 Ultra gas chromatography–mass spectrometer (GC–MS), equipped
with an AOC- 20i auto sampler and a split-splitless injector operated in splitless mode. A
Zebron ZB-5HT Inferno GC column (30 m x 0.25 mm x 0.25μm) was used for separation.
The GC oven temperature was programmed from 60–180°C at a ramp of 20°C min$^{-1}$ followed
by a ramp of 4°C min$^{-1}$ until 320°C where it was held for 20 min. MS operating conditions





were set to an ion source temperature of 200°C and 70eV ionization energy. Spectra were
collected using GC solution Workstation software (v2). *n*-alkanes, $C_{25}$ highly branched
isoprenoids (HBIs) and botryococcene compounds were identified by retention times and
comparison against mass spectra from the literature. Quantification of the *n*-alkanes, $C_{25}$ HBIs
and botryococcene compounds was done with an external standard consisting of a mixture of
$C_{20-40}$ *n*-alkanes of known concentration. Specifics on the mass spectra and retention times of
the *n*-alkanes, HBIs and Botryococcenes, including chromatograms as reference, are included
in Supplement  (Fig. S1-S6).
**2.4   δD analysis of leaf waxes**
The F1 fraction was further separated into three fractions (F1a, F1b and F1c) over a pipette
column filled with 10% $AgNO_3$-coated silica gel. F1a, which comprises *n*-alkanes, was eluted
with hexane; F1b, made up of a few unidentified compounds, was eluted with hexane-DCM
(1:1); and F1c consisting of HBIs and botryococcenes, was eluted with DCM-Acetone (9:1).
F1b and F1c were also stored in the freezer for further analysis.  F1a was analyzed by gas
chromatography–isotope ratio monitoring–mass spectrometry (GC-IRMS) using a Thermo
Finnigan Delta V Plus mass spectrometer interfaced with a Thermo Trace GC 2000 using a
GC Isolink II and Conflo IV system. Helium was used as a carrier gas at constant flow mode.
The GC oven temperature was programmed from 100–250°C at a ramp of 15°C min$^{-1}$
followed by a ramp of 10°C min$^{-1}$ until 320°C where it was held for 9 min. A standard
mixture of *n*-alkanes with a known isotopic composition (reference mixture A4, provided by
Arndt Schimmelmann, Indiana University, USA) was run several times daily to check
instrument performance and to calibrate the reference gas ($H_2$) against which the samples
were measured. All analyses were performed in triplicate and results are reported as the
weighted mean. The average standard deviation for standards and samples was around 4‰
(see Supplement).
**2.5   DNA extraction and qPCR**
Freeze-dried samples were selected according to initial biomarker screening results, in order
to estimate the abundance of different groups of organisms related to: 1) the *Prokarya*,
*Archaea* and *Bacteria*, *Cyanobacteria,* and microorganisms involved in anaerobic methane
cycling (quantification of the *mcrA* gene, e.g., Hallam et al., 2003, Hallam et al., 2004), 2)
*Eukarya*, diatoms, and *Botryococcus sp.* The samples were analyzed in order to specifically





reflect the sample conditions used for the biomarker analysis. Freeze-drying was not expected
to introduce significant biases but enhances cell breakage and the release of intracellular
DNA, following the freeze thaw method of DNA extraction (e.g. Tsai et al., 1991). This is
especially useful for soil and sediment samples (e.g. Tsai et al., 1991). Around 0.2 g (from
0.17 to 0.26 g) of freeze-dried sediment was extracted for DNA, using the MoBio PowerSoil®
DNA kit (Carlsbad, CA), following the manufacturer's instructions. About 500 µL of sterile
PBS 1X was also added to PowerSoil® Bead tube in order to enhance cell lysis efficiency.
The qPCR amplifications were conducted in 96 well qPCR plates in a CFX96 Touch™ Real-
Time PCR Detection System Instrument (C1000 Touch™ Thermal, Cycler, Bio-Rad) and its
software. The reactions consisted of a final volume of 25 µL, using the SsoAdvancedTM
Univesal SYBR® Green Supermix (Bio-Rad) following the manufacturer's recommendations.
Reactions run in 35 cycles contained 5 µL of DNA template and specific primer sets at their
appropriate concentrations and annealing temperatures (see Supplement).
Standard curves were calibrated using ten-fold serial dilutions from pure cultures of each
representative target group (Supplement). The qPCR detection of 16S rRNA genes, 18S
rRNA gene as well as *mcrA* genes in all of the samples and in ten-fold serial dilutions used to
construct the standard curves was run in triplicates. For each qPCR, several negative controls
were performed in order to check for laboratory contamination. The efficiencies of the qPCR
analyses was up to 90% with a correlation to the standard calibration curve of up to $R^2=0.996$
(see Supplement).
A total of 16S rDNA, 18S rDNA and *mcrA* gene copy numbers per g of sediment were
calculated from the triplicate average of each sample as described by (Sylvan et al., 2013).
Overall Prokaryotic cell abundance per gram of sediment was estimated by taking into
account the average of the 16S rRNA gene per cell equivalent to 1.86 for *Archaea*, 4.1 for
Bacteria (Lee et al., 2009) as previously used by (Sylvan et al., 2013) and 2.18 for
*Cyanobacteria* (after calculation of the average using the data from (Schirrmeister et al.,
2012). Due to lack of references from lake sediments, one copy per cell of the *mcrA* gene was
used to quantify the population of organisms involved in anaerobic methane cycling.
The raw data of *Eukarya*, diatoms and *Botryococcus sp.* were not further quantified into copy
numbers per g sediment, for two reasons: 1) there is high variability in 18S rRNA gene copies
per cell within the *Eukarya* and diatoms (i.e. from 3 to more than 25000 copies per cell in the
plants ((Prokopowich et al., 2003, Zhu et al., 2005) and between 61 to 36,896 for the diatoms





(Godhe et al., 2008)); 2) the paucity of information related to the number of 18S rRNA gene
copy number in the *Botryococcus sp*. genome. Therefore, the results reported here are
indicative from the universal *Eukarya* primer and should be considered as relative abundance
of the total *Eukarya* due to the tendency of not detecting all Eukaryotic groups. Yet, the data
are still useful to depict trends in the sediment record. The limitations of this method are
given in Supplement.
**3 Results**
The $^{210}$Pb activity shows an exponential decay curve with depth, which shows a decreasing
linear trend when plotted on a log-scale (i.e. ln ($^{210}$Pb$_{unsupported}$ *vs* depth; $r^2$ = 0.827) (Fig. 2).
The profile indicates minimal sediment bioturbation, and is used to calculate an average
sedimentation rate of about 4.7 mm yr$^{-1}$, which is similar to that of estuarine sediments from
the eastern coast of Thailand (Cheevaporn and Mokkongpai, 1996).
Biogeochemical and biolipid screening of the sediment core, discussed further below,
demarcates three distinct units: unit I from the top to 20 cm depth, unit II between 20 and 45
cm depth and unit III between 45 and 74 cm depth. Increasing sediment depth is linearly
related to increasing sediment age, such that unit I corresponds to ~2008-1969 CE, unit II to
~1969-1916 CE and unit III to ~1916-1857 CE.
The sediments are highly organic with TOC contents of between 30 and 40%. TOC (%)
gradually decreases with depth (units I and II), but increases sharply in the middle of unit III
and remains high until the bottom of the sequence (Fig. 3a). Both $\delta^{13}C_{bulk\ org}$ (Fig. 3b) and
total N (%) (Fig. 3c) show a gradually decreasing trend downcore while $\delta^{15}N_{org}$ values
increase in unit I and then steadily decrease through units II and III (Fig. 3d). The C/N ratio
on the other hand increases gradually from the top to the bottom of the sediment core (Fig.
3e). Si/Ti and O/Ti ratios, markers of nutrient input into the lake from terrestrial sources,
show strong co-variation and general decrease downcore, with the highest ratios recorded in
unit II (Figs. 3f and g). The covariation between Si/Ti and O/Ti ratios suggests that SiO$_2$
dominates among the silicate minerals in the catchment. Pictures taken using ESEM show
higher abundances and diversity of diatoms in unit II compared to units I and III (see
Supplement; Fig. S7). The P/Ti ratio, which can be used as a proxy for eutrophic conditions in
lakes (Kirilova et al., 2011), decreases with depth (Fig. 3h).



Botryococcene lipid concentrations have low values in unit I, increase in relative abundance
in unit II and are most abundant in unit III, with a maximum at ~60 cm depth (~1880 CE)
(Fig. 4a). HBIs, which have very low relative abundances in unit I, sharply rise in unit II
where they maximize at ~30 cm (~1950s), and exhibit intermediate and slowly decreasing
concentrations in unit III (Fig. 4b). The relative abundance of $C_{17}$ $n$-alkanes is markedly high
in especially the top of unit I, but is low throughout units II and III (Fig. 4c). The hydrogen
isotopic composition of leaf waxes ($\delta D_{wax}$; weighted mean of $\delta D$ $C_{27-31}$ $n$-alkanes) shows a
long-term oscillation over the entire 150-year record (Fig. 4d). The most recent decades
exhibit an increasing (drying) trend, when compared to the values of unit II. Minimum values
(wettest conditions) of $\delta D_{wax}$ are reached in the middle of unit II. Relatively high values
(driest conditions) are found halfway through unit III (around 60 cm depth, ~1890 CE), after
which $\delta D_{wax}$ values slowly decrease, suggesting a hydroclimatic trend towards relatively wet
conditions.
The qPCR data set shows generally identical trends for total Prokaryotes, *Eukarya* and
*Botryococcus sp.* (Figs. 5a-c). Their abundances decrease with depth (until about 43 cm) and
then increase gradually again in unit II, except for the numbers of the prokaryotes, which in
addition to the general trend also show a sharp spike in abundance at around 30 cm depth.
Bacterial abundance displays a generally decreasing trend without delineated structure (Fig.
5d), whereas the *Cyanobacteria* numbers (presented as a percentage of the Bacteria
quantified) decrease sharply with depth in unit I and then remain relatively low throughout the
entire sequence (Fig. 5e). Diatom abundance increases gradually in unit I and then sharply in
unit II before dropping significantly to a minimal level at the transition between unit II and III
(Fig. 5f). *Archaea* abundance only shows minimal variations (Fig. 5g). Bacterial communities
dominate among the prokaryotes throughout the whole sediment core (attaining more than
90% of the total prokaryotic abundance). *mcrA* genes were mostly detected in the upper part
of the sequence where they represent up to 8.5% of the *Archaea*, but also were found in
substantial amounts at depths of 26 and 61 cm. The *mcrA* gene has a maximum occurrence in
units I and III and relatively low abundance in unit II (Fig. 5h). The presence of *mcrA* genes
in the sediment likely indicates anaerobic methane cycling processes (anaerobic methane
oxidation and methanogenesis) (Hallam *et al.,* 2004) and this shows a significant correlation
with total *Eukarya* (Fig. 6; $r^2 = 0.85$).





High proportions of *Cyanobacteria* (>8-26% of the total *Bacteria* numbers) and a low diatom
biomass (~1-1.4% of the total *Eukarya* abundances) characterize unit I. In contrast a dense
diatom signature (up to 1.7% of the total *Eukarya* abundances) and low proportions of
*Cyanobacteria* (~1% of the total bacteria numbers) is distinctive for unit II. *Cyanobacteria*
*and* diatom abundances are however relatively low in unit III. The transition between unit II
and III is marked by an important decrease in total biomass, which was characterized by low
estimates of eukaryotic and prokaryotic numbers (Fig. 5). *Botryococcus sp.* abundances show
an increase in unit I and unit III on a log scale, which correlates with the concentration of
botryococcenes, except for unit I, suggesting that in unit III most of the *Botryococcus sp.*
detected by qPCR could be *Botryococcus braunii*.

## 4   Discussion

### 4.1   Carbon cycle in NPK

The biogeochemical trends suggest that multiple processes control the organic matter (OM)
input into lake NTP, which in turn play a significant role in carbon storage. Increasing C/N
ratio with depth (Fig. 3e) may be explained by a preferential (anoxic) mineralization of OM
rich in N leading to residual OM with a higher C/N ratio, as observed in many other systems
(Emerson and Hedges, 2003). This explanation is also consistent with long-term diagenetic
OM transformation (Sun et al., 2004). Successional deposition of different phytoplankton
communities with different C/N ratios can also explain the C and N signal, which could
represent an original signal of deposition: replacement of lipid-rich *Botryococcus* with
particularly high C/N ratio in unit III by diatoms as the dominant plankton around 1920,
which were in turn replaced by cyanobacterial activity in the second half of the last century,
with low C/N ratio due to their capacity to fix N. Alternatively, the pattern of decreasing
$\delta^{13}C_{org}$ (Fig. 3b) while TOC increases (Figs. 3a) with depth could also be attributed to
successional shifts of trophic state of the lake (Brenner et al., 2000; Meyers and Teranes,
2001). Increase in nutrients from one trophic state to the other decreases the amount of
dissolved $CO_2$ available for use by the phytoplankton community. This leads to lower net
fractionation against $^{13}C$ by the phytoplankton during photosynthesis and thus increasing $\delta^{13}C$
values (Meyers and Teranes, 2001).
The presence of *mcrA* genes in the sediment can be a remnant signal of past methane cycle
activity (Hallam et al., 2004) in the upper sediment and/or the anoxic bottom waters of the





lake but can also represent ongoing methanogenesis within the sediment (Stein et al., 2001;
Earl et al., 2003). During methanogenesis, degassing of $^{12}$C-enriched methane could lead to
enriched $^{13}$C in residual organics (Ogrinc et al., 2002). A strong correlation between *mcrA*
gene abundance and *Eukarya* (Fig. 6; $r^2 = 0.85$) could indicate that the depth profiles reflect a
concurrency of primary productivity and methane cycling in the anoxic lake bottom waters. It
is, however, also possible that methane cyclers in the lake sediments are living off the organic
matter deposited by the phytoplankton community in the lake surface.

## 4.2 Climate influence on lake evolution and phytoplankton community changes

The combination of all analyzed proxies (biolipids, $\delta D_{wax}$, qPCR, bulk CN isotopes, and
sedimentary geochemistry) allows discussing microbial community changes in NTP and
further constrains the parameter(s) that caused the shifts in lake status through time. Decadal
changes in NTP trophic states were accompanied by variations of dominant phytoplankton
community. The period from ~1857 to 1916 CE, is marked by significant increases of both
botryococcene lipids (Fig. 4a) and *Botryococcus sp.* abundance (Fig. 5c), which also
corresponds with lower precipitation (Fig. 4d). Several studies have shown that these algae
are tolerant to oligotrophic conditions (Souza et al., 2008) and can therefore be used as a
proxy for oligotrophic lake water conditions in the oxygenated epilimnion of the lake
(Waldmann et al., 2014). The presence of the *mcrA* gene in appreciable amounts indicates
substantial microbial activity by anaerobic microbial methane cyclers in the anoxic bottom
waters and/or sediment feeding off primary producers (*Botryococcus sp.*). Our interpretation
for this is one of a fairly strongly stratified lake where reformed nutrients stayed in the anoxic
hypolimnion thereby keeping the surface water oligotrophic. In the tropics, the mean air
temperature (MAT) is a direct result of incoming solar radiation and the relationship between
the MAT and the amount of rainfall are typically inversely proportional (Imboden and Wüest,
1995; Boehrer and Schultze, 2008). Dry, cloudless and warmer conditions lead to stronger
stratification in fresh water lakes (Imboden and Wüest, 1995).
A stark difference in the dominant phytoplankton community is observed from ~1916-1969
CE. This period is marked by significant decrease in botryococcene lipid concentrations  (Fig.
4a) and *Botryococcus sp.* gene abundance (Fig. 5c), and an increase in both diatom abundance
(Fig. 5f) and $C_{25}$ HBI concentrations (Fig. 4b), which is a useful indicator of diatom-derived
OM  inputs  to  sediments  (McKirdya  et  al.,  2013).  Diatoms  dominate  phytoplankton



communities as long as there is abundant silica irrespective of changes in environmental
conditions and nutrient levels (Egge and Aksnes, 1992). Interestingly, the increase in diatom
markers (Fig. 4b and Fig. 5f) coincides with an increase in reconstructed rainfall intensity
(Fig. 4d). Moreover the increase in Si/Ti ratios (Fig. 3f), a run-off signal (Murphy et al.,
2000), coincides with high diatom blooms especially in unit II. Since Ti is a highly immobile
element, weathering and transportation of Si is not accompanied by significant Ti delivery to
aquatic basins. Therefore the Si/Ti ratio can serve as a proxy for nutrient dynamics linked to
hydrological changes (Cartapanis et al., 2014) and as an indicator for enhanced diatom
production in lakes (Wennrich et al., 2014). Altogether, it appears that the generally wetter
conditions between ~1916 and 1969 CE increased catchment runoff into the lake. The
catchment runoff in turn increased the nutrient and silicate mineral content of the lake water
(e.g. Paerl et al., 2006), changing it from oligotrophic to mesotrophic. Increased diatom
diversity and fluvial deposits observed from the image scans of the sediments further
substantiate the hydrologically driven diatom blooms (Supplement; Fig. S7). In addition, an
increase in precipitation was likely accompanied by cooler temperatures, as explained above,
which lead to a decrease in stratification and to an increase in mixing between the epilimnion
and hypolimnion. Risen nutrient levels available for the phytoplankton community may also
have been due to the mixing processes.
Between ~1969 and ~2008 CE, the phytoplankton community structure changed again, with a
diminishing role for diatoms as evidenced by lower concentrations of $C_{25}$ HBIs (Fig. 4b) and
the start of a marked increase of *Cyanobacteria* gene numbers (Fig. 5e) and $C_{17}$ *n*-alkanes
(Fig. 4c). $C_{17}$ *n*-alkanes are recognized biomarkers of aquatic algae and photosynthetic
bacteria such as *Cyanobacteria* (Meyers, 2003). Indeed, *Cyanobacteria* gene numbers relative
to Bacteria quantification based on the qPCR data covary strongly with $C_{17}$ *n*-alkane
concentration, which confirms that the $C_{17}$ *n*-alkanes were produced mainly by
*Cyanobacteria*. Additionally, the decreasing $\delta^{15}N_{bulk\ org}$ values during this period (Fig. 3d) also
suggest intense nitrogen fixation, a process strongly associated with *Cyanobacteria* activity
(Vahtera et al., 2007). This time period (~1969-2008 CE) is also characterized by somewhat
lower rainfall amounts (Fig. 4d). The amount of Si indeed decreased and diatoms became less
abundant, allowing non-diatomaceous phytoplankton not dependent on Si to take over (Egge
and Aksnes, 1992), while the amount of *Botryococcus* genes increased again (Fig. 5c). The
amount of *mcrA* genes, indicating a stronger methane cycling, also showed higher levels (Fig.
5h). However, this was not accompanied by the production of botryococcene lipids,





suggesting that another strain of these green algae became prevalent. Indeed, *Botryococcus*
*braunii* are classified into three main races: A, B and C and it is only the B race that produces
botryococcenes lipids (Eroglu et al., 2011).
The strong *Cyanobacteria* prevalence suggests a eutrophic phosphorous-rich condition
instead of the oligotrophic conditions occurring a century earlier and this notion is supported
by higher level of P in the sediment (Fig. 3h). P bioavailability is one of the most important
factors limiting aquatic *Cyanobacteria* blooms (Paerl and Fulton, 2006; Paerl and Valerie,
2012). The source of the elevated phosphorus is unclear, but likely results from human
activities. Under a land development program in the 1990s, more than 20% of Thailand's
56,000 villages were located within forest reserves (Gray, 1991; Puri, 2006), which allowed
the expansion of land encroachment and agricultural activities. For instance, southern
Thailand has seen an increase in the cultivation of rubber trees on small farms at rates above
7% yr$^{-1}$ (Leturque and Swiggings, 2011). The use of fertilizers in farming activities, untreated
wastewater effluents and the use of detergents are likely sources of the elevated phosphorus
inputs into the lake (Litke, 1999; Chislock et al., 2013). These accelerated the eutrophic state
of the lake beyond the natural rate of nutrient enrichment, which takes centuries to achieve
(Litke, 1999).
According to our analysis, photosynthetic primary production is at the base of internal organic
matter production in the lake. However, changes in precipitation, anthropogenic forcing and
nutrient input have produced fluctuations in dominant primary producer communities over the
last 150 years, from botryococcene lipid-producing algae to diatoms and then currently to
*Cyanobacteria* predominance.
**5   Summary and conclusion**
The combination of geochemistry, lipid biomarker and qPCR analyses allowed distinguishing
and quantifying different microbial groups in the sediments of Lake Nong Thale Prong,
southern Thailand, and most importantly allowed the identification of biological relationships
between the phytoplankton community structure response to either natural environmental
changes or anthropogenic impact.
Between ~1857 and 1916 CE, relatively drier climate in southern Thailand coincided with
oligotrophic surface water conditions in Lake NTP, which was dominated by botryococcene
lipid-producing primary producers. These likely sustained anaerobic methane cycling in the





anoxic bottom waters and sediment, as evidenced by the detection of *mcrA* genes. A change
to higher Si input into the lake could be linked to increased precipitation from ~1916-1969
CE, which led to a rapid takeover by diatoms as primary producers. The increase in
precipitation was likely accompanied by decreased stratification with a greater mixing of
reformed nutrients from the depth to the surface water and a decrease of methane cycling
related genes. Since the 1970s many aspects of the initial limnic state returned upon drier
conditions, except that anthropogenic impact led to an increase in P allowing cyanobacteria to
become an important contributor to primary productivity.
The change in TOC values and C/N ratios, which decrease between ~1857 CE and present,
suggest that the botryococcene lipid-producing *Botryococcus braunii* were a key part of an
alternate stable lake status that facilitated the most efficient capture and burial of C in the
sediments (~1857-1916 CE). The *mcrA* gene abundance suggests strong anaerobic methane
cycle dependence on the primary producers, phytoplankton community. However, processes
like lake stratification and mixing between the epilimnion and hypolimnion possibly affected
the *mcrA* gene abundance: strong stratification leads to increase in *mcrA* gene abundances
(units 1 and III) whereas mixing leads to decrease in *mcrA* gene abundances.
Our results show that the combinations of biolipid analysis, $\delta D_{wax}$, qPCR, bulk isotopes of C
and N, and sedimentary geochemistry are effective in unraveling how external forcing
(hydroclimate and human impact) influences internal abiotic feedback processes. The abiotic
feedback processes as a result of changing climate has implications for phytoplankton regime
shifts and their role in carbon capture storage and suggest that phytoplankton sedimentary
records may assist in tracking such changes over decadal to centennial timescales.

**Author contributions**

This study was conceived and led by K. A. Yamoah, E. Chi Fru and R. H. Smittenberg. K. A.
Yamoah, N. Callac, A. Wiech and A. Chabangborn carried out laboratory analyses. K. A.
Yamoah, N. Callac, E. Chi Fru, B. Wohlfarth and R. H. Smittenberg wrote the manuscript.
All authors discussed the results and their implications and commented on the manuscript as it
progressed.

**Acknowledgements**

This work was supported by Swedish Research Council (VR) research grants 621-2008-2855,
348-2008-6071 and 621-2011-4916 to Barbara Wohlfarth and Rienk Smittenberg. We wish to




thank Sherilyn Fritz, Wichuratree Klubseang, Sudo Inthonkaew, Minna Väliranta and
Sakonvan Chawchai for sampling assistance; Jayne Rattray, Anna Hägglund, and Christoffer
Hemmingsson for laboratory assistance; Alfred Burian for providing some eukarial strains;
Anna Neubeck for providing the methanogen strain; and Frederik Schenk for assisting with
observational precipitation data.

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

29



1 **Figures**

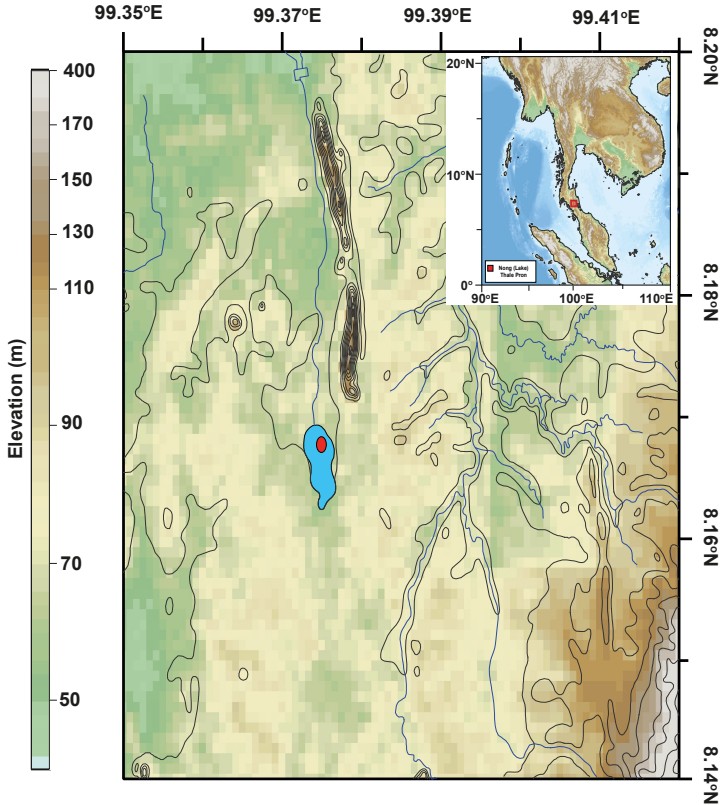

Figure 1. Location of the study area in southern Thailand and topography of Lake Nong Thale
Prong (shaded blue). A red circle shows the coring site. For interpretation of the references to
color, the reader is referred to the web version of this article



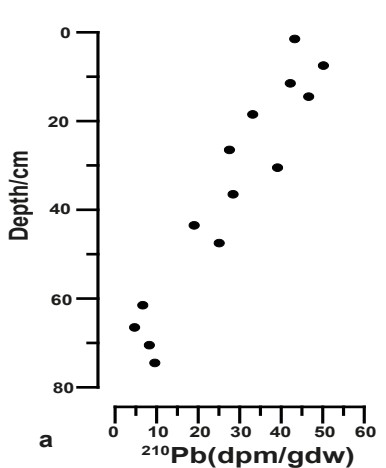

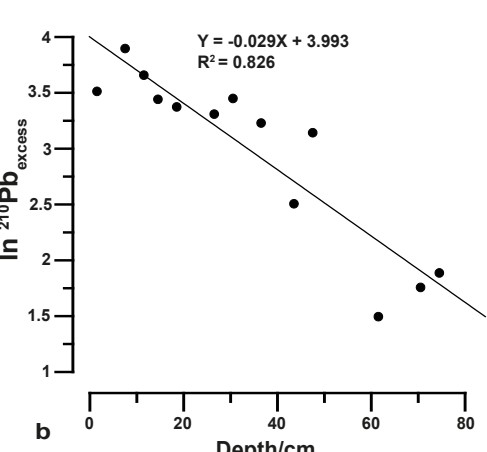

Figure 2. Variations in $^{210}$Pb down the sediment core, (a) Depth profile of total $^{210}$Pb activity

downcore and (b) Correlation between depth and ln $^{210}$Pb$_{excess}$.





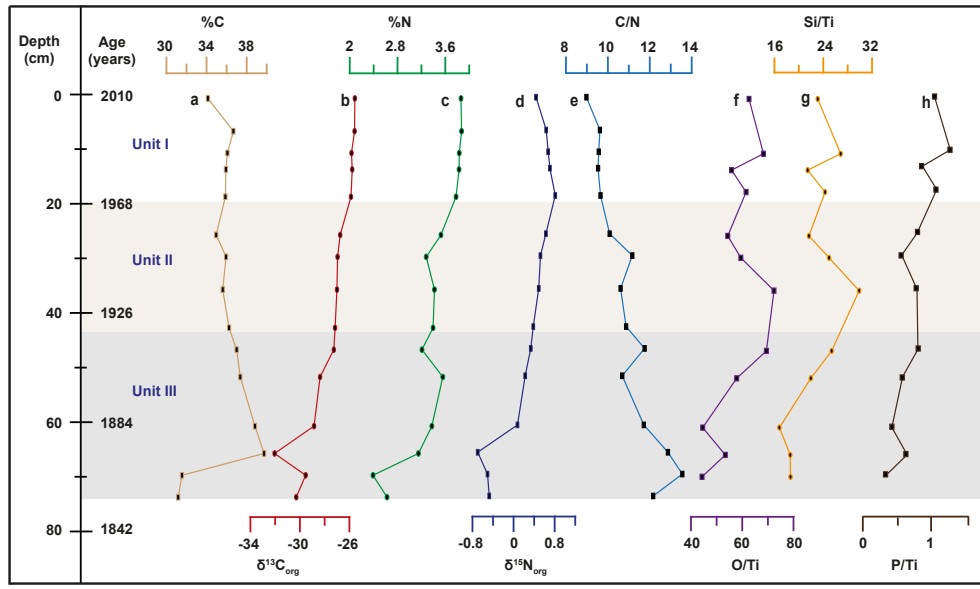

Figure 3. Geochemical data from Lake Nong Thale Pron (NTP) plotted against depth and age
(a) TOC (%), (b) $\delta^{13}C_{bulk\ org}$, (c) TN (%), (d) $\delta^{15}N_{bulk\ org}$, (e) C/N, (f) Si/Ti, (g) O/Ti, (h) P/Ti.
The shaded boxes represent the transition between the different units I, II and III. For
interpretation of the references to color, the reader is referred to the web version of this
article.



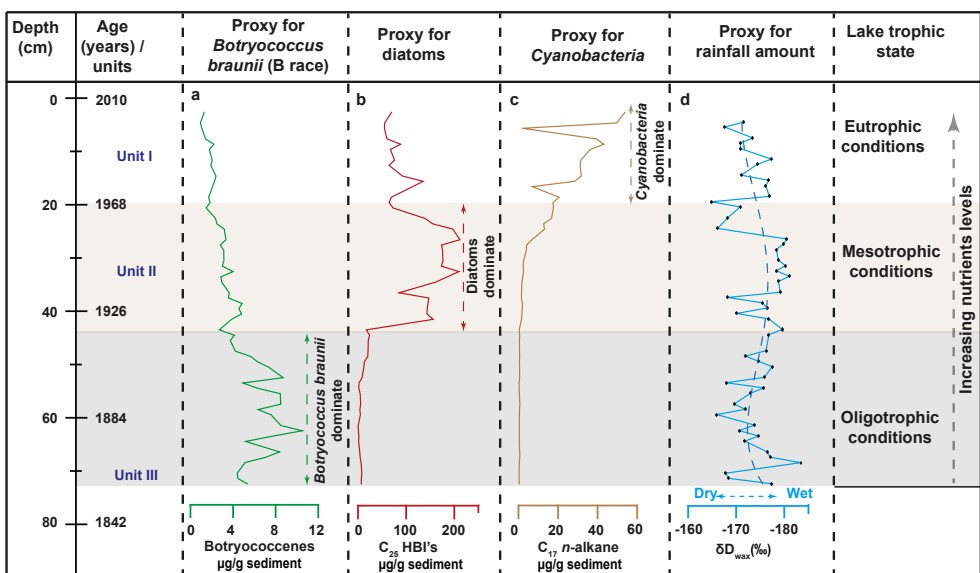

Figure 4. Depth and age profiles of lipid biomarkers, (a) botryococcenes, a proxy for *Botryococcus braunii* (B race) (b) $C_{25}$ Highly branched Isoprenoid (HBIs), a proxy for diatoms (c) $C_{17}$ *n*-alkane, a proxy for *Cyanobacteria* and (d) δD of $C_{27-29-31}$ *n*-alkanes (δD_wax), a proxy for rainfall amount. The blue line through the δD_wax data set represents a polynomial fitted trendline and the shaded boxes represent the transition between units I, II and III. The last column shows the different trophic changes with time.



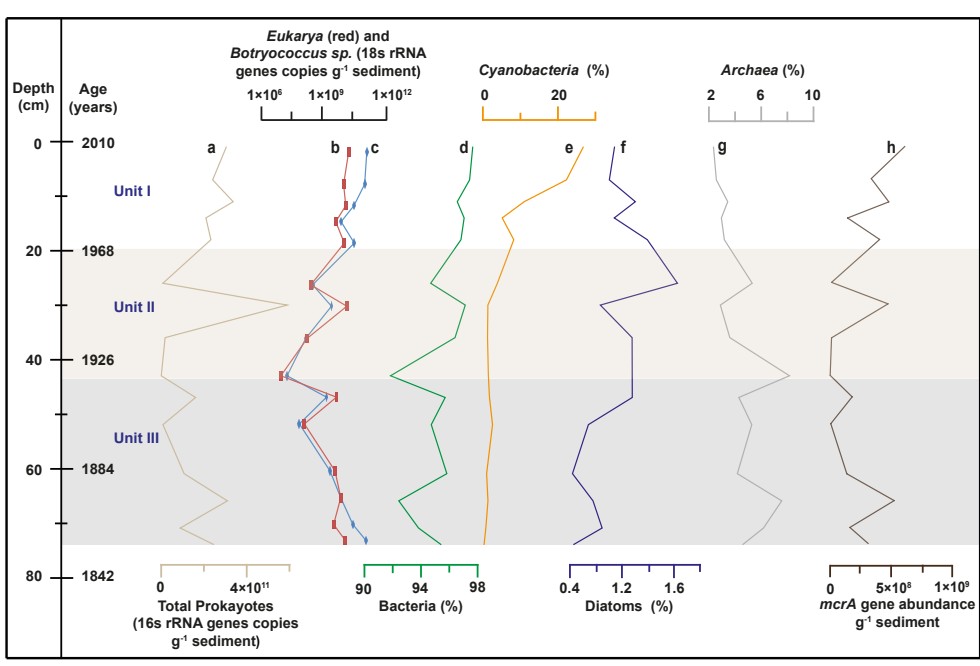

Figure 5. Depth and age profiles for: (a) Total prokaryotes (16s rRNA genes copies g$^{-1}$ sediment), (b) *Eukarya* (18s rRNA genes copies g$^{-1}$ sediment), (c) *Botryococcus sp.* (18s rRNA genes copies g$^{-1}$ sediment), (d) Bacteria (%), (e) *Cyanobacteria* (%), (f) Diatoms (%), (g) *Archaea* (%), (h) *mcrA* gene abundance g$^{-1}$ sediment. The shaded boxes represent the transition between units I, II and III

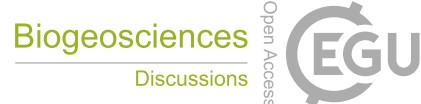



10    Figure 6. Cross plot between *mcrA* gene abundance against *Eukarya* as a proxy for total

11    primary productivity

