# Peer review of "A 150-year record of phytoplankton community succession controlled by hydroclimatic variability in a tropical lake"

_Biogeosciences, 2015_

## Referee Comment (RC1) · Anonymous Referee #1 · 14 Feb 2016

General comments: The manuscript by K. A. Yamoah et al. presents phytoplankton community succession and geochemical variations over the past 150 years in a tropical lake in southern Thailand. Overall, the authors comprehensively collected DNA, lipid, elemental data, and drew relevant conclusions. But, there are specific and technical problems that should be resolved, so I think that the manuscript doesn't meet the requirements for publication on GB.

Specific comments: The authors concluded that hydroclimate change and anthropogenic activities played an important role in phytoplankton succession. However, the authors just mentioned "hydroclimate" in the title, but I suggest "anthropogenic activities" should be also included in the title.
[Figure]

Some more specific comments as follows: Page 3, line 18-19: Please specify "external" and "internal" ecosystem regulators. Page 4, line 19: it may be better to change "chemical environment" to "chemical and physical environment". Page 10, line 9-13, authors show wet/dry conditions in parenthesis. Pls explain how the results "wet/dry condition" were inferred, and include appropriate inferences. Page 11, line 3, change "Eukarya" to "eukaryotic". Page 11, line 16-17, "as observed in many other systems (Emerson and Hedges, 2003)", two or more references should be cited here. Page 12, line 2-3: references cited here suggested that 13C enriched is in residual organics. However, in Unit III, $\delta$13C was more negative, while mcrA abundance was relatively high. Please explain the inconsistency. Page 12, line 4-5, the sentence is obscure, i.e. "eukarya" doesn't represent all "primary productivity", which includes both cyanobacteria and eukaryotic algae. Please clarify it. Page 12, line 20, does "microbial activity by anaerobic microbial methane cyclers" mean "methanogensis"? Page 13, line 7-8: Cartapanis et al. 2014 used opal other than total Si elemental concentration to infer nutrient dynamics. I'm not sure if it is appropriate to use Si concentration in this study. Page 13, line 25, "which confirms that the C17 n-alkanes were produced mainly by Cyanobacteria" seems too arbitrary. I suggest to change it to "which suggested that the C17 n-alkanes may be produced mainly by cyanobacteria" Page 14, line 2, it's better to replace "race" with "lineage" or "subgroup". Page 14, line 8, "likely results" should be "is likely resulted". Page 14, line 10, replace "within" with "in". Page 14, line 13, replace "in" with "during". Paragraphs within "Summary and conclusion" from page 14, line 14 to page 15, line 16 are just a repeat from the last section. I suggest that these sentences should be deleted.

Technical corrections: A lot of terms should not be italic or capitalized. For examples Bacteria, Cyanobacteria, Eukarya, sp. Pls check. Page 3, line 22-25, the sentence is confusing. Please revise it. Change "factors that" to "which". Page 12, line 22-23, the sentence is hard to understand. Pls rewrite it. Page 13, line 23, "photosynthetic bacteria such as Cyanobacteria" can be changed into "cyanobacteria". Page 11, line 3, change "Eukarya" to "eukaryotic". Page 11, line 5, "and" should not be italic.

---

## Referee Comment (RC2) · Anonymous Referee #2 · 21 Mar 2016

Review comments for Yamoah et al., A 150-year record of phytoplankton community succession controlled by hydroclimatic variability in a tropical lake. (MS No. bg-2015-633)

The authors presented a data set of lipids abundances, compound specific hydrogen isotope, bulk carbon and nitrogen isotopes, and DNA from a sediment core, to investigate decadal variations in phytoplankton communities in a ∼150 year of a tropical lake. Although the authors make an effort to establish a new methodology to evaluate the ecological changes in the lake by these biological and chemical analyses, this paper is rather descriptive and spotty discussion, and lacks the in-depth discussion. As long as the authors presented a lot of data set, I think that the authors should comprehensively

discuss the lake environment rather than picking up the specified topic. Therefore, in my opinion, a substantial revision is required to make this MS suitable for publication in BG.

Detailed comments: Page 3, line 11: Meaning of the terms "external" and "internal" ecosystem should be specified. Page 5, line 14: Please describe the depth (m) of the sampling location, the "deepest part". Page 5, lines 19-22: I suggest to tone down this part. Robador et al 2015 did not give you off-handed support nor the guarantee for storing sampled sediments without froze them for days. Organic compounds and its isotope compositions can very likely be affected. "Such" temperatures needs to be specified. page 5, line 22: It would be better to remove the word "biogeochemical". Page 6, lines 3-5: Descriptions of the standard materials for carbon and nitrogen isotope analysis (e.g. working standards) should be included. Page 6, lines 16-29: It will be better to cite original papers for the method. Page 7, lines 10-25: For the delta-D analysis, please present at least one set of IRMS chromatogram from analyzed sample. For the compound specific isotope analysis, especially for the $\delta$D, single-peak baseline separation of targeted compound is essential to get reliable data. It will be better to cite original papers for the analytical method. Page 9, line 10-12: Specify the reason to compare sedimentation rates between a lake and a near -by estuary? The two aquatic fields have completely different physical natures, I failed to see the reason or the necessity for their sedimentation rates should be in the same range. Page11 Sec.4.1: The description of the data trend is rather difficult to understand. Please indicate specific unit name or age from each figures when discussing. Unclear description made it difficult to follow the thread of your discussion. (e.g. p.11 line 22, "the second half of the last century in Figure 3" can be addressed by year) Page 11, lines 17-23: This part fails to convince the readers, as some of the discussion seems to be contradictive. I think to drew this conclusion, the d15N variation in surrounding watersheds, substrate nitrogen, actual values of phytoplankton and N-fixing cyanobacteria should be considered and discussed. Especially, when the lake is small and easily affected by surrounding environments. The same thing can be said about the discussion regarding

d13C trends. The reason why the low C/N ratio can be the direct indication to the sift in dominant plankton from diatoms to cyanobacteria should be addressed, too. Page 12 Sec.4.2: Recently, Chikaraishi et al. (2012) reported that the terrestrial insects have long-chain n-alkanes (C21–33) with lighter $\delta$D (-195 $\pm$ 16‰ abundantly. Does this affect some of your discussion in this section, as the contribution of insect-derived n-alkanes can be one of the reasons for the negative sift in $\delta$D records? Supplement page5 sec 4.3: The English is difficult to understand. Table S1: Please address units for S, O2, P. Figure S1: Captions should explain all the symbols or lines in the figure. Please remove excess notes.

---

## Author Response (AR1)

Stockholm University
Department of Geological Sciences
Stockholm University
10691 Stockholm, Sweden
+46 (0) 8167832, 8164760
kweku.yamoah@geo.su.se

The editor,
Biogeosciences

Stockholm, 1 June 2016

Dear editor,

We hereby submit a revised version of our research article titled "*A 150-year record of phytoplankton community succession controlled by hydroclimatic variability in a tropical lake*", by Yamoah K. K. Afrifa, Nolwenn Callac, Ernest Chi Fru, Alan Wiech, Barbara Wohlfarth, Akkaneewut Chabangborn and Rienk H. Smittenberg.

The manuscript has been carefully revised and all the comments raised by the reviewers have accordingly been addressed. In addition, we have also updated and modified most parts of the discussion and conclusion, especially the interpretations related to the *mcrA* genes abundance and productivity. These changes however do not affect the core interpretation of the data presented in this manuscript. I have therefore attached another manuscript with highlighted (yellow) changes for your perusal.

If you have any questions please do not hesitate to contact me.

Sincerely,

Yamoah Kweku Kyei Afrifa

**General comments:**

*The manuscript by K. A. Yamoah et al. presents phytoplankton community succession and geochemical variations over the past 150 years in a tropical lake in southern Thailand. Overall, the authors comprehensively collected DNA, lipid, elemental data, and drew relevant conclusions. But, there are specific and technical problems that should be resolved, so I think that the manuscript doesn't meet the requirements for publication on GB.*

Answer

We sincerely thank the reviewer for taking the time to review our manuscript. The comments and suggestions made have been taken into account in the revised manuscript. We respectfully disagree with the particular comment that our manuscript does not meet the requirements for publication in Biogeosciences based on technical problems. Below we provide justifications to enable the right editorial decision to be made.

*Specific comments: The authors concluded that hydroclimate change and anthropogenic activities played an important role in phytoplankton succession. However, the authors just mentioned "hydroclimate" in the title, but I suggest "anthropogenic activities" should be also included in the title.*

Answer

Although anthropogenic activity does play a role over the last recent 40 years with increased phosphorus levels, this is not the main focus of the paper. Instead, we highlight the influence of natural hydroclimate variability on phytoplankton community change over a ~150 year period. The main important factor why hydroclimate variability is highlighted in the title is because, without the changes in hydroclimate conditions, new resources entering the lake will be limited, including Si and P input, regardless of the source. Particularly, the atmospheric component of P is weak and continental weathering and runoff transportation often promote supply to aquatic systems. That is to say that the degree of drainage into the lake controlled the phytoplankton shifts, not anthropogenic activity. We have identified two key changes in nutrient input sources that are directly coupled to hydroclimatic dynamics: (1)

changes in Si input originating from weathering of local rocks and (2) phosphorus originating from anthropogenic sources, which have been brought into the lake by runoff intensity. Therefore the title captures the main pathway to nutrient input into the lake, not the processes that generate the nutrients, namely weathering and anthropogenic sources. Otherwise we will also have to change the title to include weathering.

*Some more specific comments as follows: Page 3, line 18-19: Please specify "external" and "internal" ecosystem regulators.*

We have changed the text to elaborate more on the external and internal factors we allude to. By external factors, we mean processes such as rainfall and anthropogenic activities that affect the lake, including weathering and runoff intensity. Internal factors include, the cycling of various nutrients within the lake, including nutrient regeneration, rates of primary productivity and organic carbon and nitrogen cycling, etc. These are factors, which we subsequently address by the amount of data provided.

*Page 4, line 19: it may be better to change "chemical environment" to "chemical and physical environment".*

We have changed this in the text.

*Page 10, line 9-13, authors show wet/dry conditions in parenthesis. Pls explain how the results "wet/dry condition" were inferred, and include appropriate inferences.*

δDwax is commonly used as a paleoclimate proxy to reconstruct moisture availability in monsoon regions, as supported by references in the manuscript and the supplementary (see page 4 line 23-25). We do not deem it necessary to discuss how the proxy works in the main text and have therefore provided citations to this effect. Rather, a comprehensive summary about the δDwax as a proxy for hydroclimate is provided in the supplementary, which we referred readers to. In addition, we provided rainfall data for much of the studied interval and compared it with the δDwax data.

Therefore we use wet and dry to clearly define more and less rainfall, respectively as alluded to in the text.

*Page 11, line 3, change "Eukarya" to "eukaryotic". Page 11, line 16-17, "as observed in many other systems (Emerson and Hedges, 2003)", two or more references should be cited here.*

We have changed the text and added the following references: Ostrom et al 1997; Altabet, 1998; Sachs and Repeta, 1999).

*Page 12, line 2-3: references cited here suggested that 13C enriched is in residual organics. However, in Unit III, δ13C was more negative, while mcrA abundance was relatively high. Please explain the inconsistency.*

We agree partly with the reviewer that this appears inconsistent and have made changes in the text to address this point better. We like to note that *mcrA* abundance was used sparingly to connect lake productivity to anaerobic methane cyclers. However, the mcrA gene is a proxy for both anaerobic methanotrophy and methanogenesis. High rates of anaerobic methanotrophy and methanogenesis tend to produce extremely negative d13Corg that can even reach -60‰ Therefore, the coincidental transition to more negative d13Corg values coupled to increasing abundance of the mcrA gene, do indeed suggest increase in either anaerobic methane oxidation of methane or methanogenesis. Therefore we have now discussed the data accordingly in the manuscript, since it appears that specifically in unit III anaerobic methane cycling might have contributed significantly to the d13Corg signature of residual organic carbon.

*Page 12, line 4-5, the sentence is obscure, i.e. "eukarya" doesn't represent all "primary productivity", which includes both cyanobacteria and eukaryotic algae. Please clarify it.*

This has been clarified in the revised text. Now it reads "Eukaryotes contribute significantly to primary production in lake systems, thus a strong correlation between mcrA gene abundance and Eukarya (Fig. 6; r2 = 0.85) could indicate that the depth profiles reflect a concurrency of primary productivity and methane cycling in the anoxic lake bottom waters."

*Page 12, line 20, does "microbial activity by anaerobic microbial methane cyclers" mean "methanogensis"?*

No, because *mcrA* genes are for anaerobic methane cycler's i.e. methanogenesis + anaerobic methane oxidation, as explain in a comment above.

*Page 13, line 7-8: Cartapanis et al. 2014 used opal other than total Si elemental concentration to infer nutrient dynamics. I'm not sure if it is appropriate to use Si concentration in this study.*

We respectfully disagree here with the reviewer. Cartapanis et al. (2014) used elemental ratios of Si/Ti as a proxy for opal, which is a hydrated amorphous form of silica. Indeed, we also used Si/Ti as a "proxy for nutrient dynamics linked to hydrological changes (Cartapanis et al., 2014) and as an indicator for enhanced diatom production in lakes (Wennrich et al., 2014)" (refer to page 13 line 7). What is of importance in our data is not the specific Si mineral in the sediments, since what form of Si remaining in the sediments reflects diagenetic and recrystallization processes. What we are interested in is mapping changes in the Si budget as a function of detrital input. Ti is a common detrital input signal. Essentially, supply of dissolved Si by runoff should vary accordingly with the Si/Ti ratio, since Ti is broadly immobile. An increase in the Si/Ti ratio implies more input and decreasing Si/Ti implies reduction in runoff supply, since the authigenic Si content of the basin is not amplified by an external source. Our approach is consistent with that used in many Paleo-ennvironmental studies and the mineral form of Si in the sedimentary basin is inconsequential. In fact, most mobile elements are often normalized to Ti to show changes in sedimentary inputs, from lake to marine systems. See for example:

1. Konhauser KO, *et al.* (2011) Aerobic bacterial pyrite oxidation and acid rock drainage during the Great Oxidation Event. *Nature* 478:369–373.

2. Mathur R, *et al.*, (2004) Cu isotopes and concentrations during weathering of black shale of the Marcellus Formation, Huntingdon County, Pennsylvania (USA). *Chem Geol* 304-305: 175–184.

3. Demory F, Oberhänsli H, Nowaczyk NR, Gottschalk M, Wirth R, Naumann R (2005) Detrital input and early diagenesis in sediments from Lake Baikal revealed by rock magnetism. *Global Plan Change*, 461:145-166.

*Page 13, line 25, "which confirms that the C17 n-alkanes were produced mainly by Cyanobacteria" seems too arbitrary. I suggest to change it to "which suggested that the C17 n-alkanes may be produced mainly by cyanobacteria"*

We agree with the reviewer. This has been changed in the revised manuscript.

*Page 14, line 2, it's better to replace "race" with "lineage" or "subgroup".*
We have changed the text according to the suggestion.

*Page 14, line 8, "likely results" should be "is likely resulted". Page 14, line 10, replace "within" with "in".*

We have changed the text as suggested

*Page 14, line 13, replace "in" with "during". Paragraphs within "Summary and conclusion" from page 14, line 14 to page 15, line 16 are just a repeat from the last section. I suggest that these sentences should be deleted.*

We have revised the text and have incorporated the reviewer's concerns.

*Technical corrections: A lot of terms should not be italic or capitalized. For examples Bacteria, Cyanobacteria, Eukarya, sp. Pls check. Page 3, line 22-25, the sentence is confusing. Please revise it. Change "factors that" to "which". Page 12, line 22-23, the sentence is hard to understand. Pls rewrite it. Page 13, line 23,*

*"photosynthetic bacteria such as Cyanobacteria" can be changed into "cyanobacteria". Page 11, line 3, change "Eukarya" to "eukaryotic". Page 11, line 5, "and" should not be italic*

We have revised the text and have incorporated the reviewer's concerns.

**Anonymous Referee #2**

The authors presented a data set of lipids abundances, compound specific hydrogen isotope, bulk carbon and nitrogen isotopes, and DNA from a sediment core, to investigate decadal variations in phytoplankton communities in a ~150 year of a tropical lake. Although the authors make an effort to establish a new methodology to evaluate the ecological changes in the lake by these biological and chemical analyses, this paper is rather descriptive and spotty discussion, and lacks the in-depth discussion. As long as the authors presented a lot of data set, I think that the authors should comprehensively discuss the lake environment rather than picking up the specified topic. Therefore, in my opinion, a substantial revision is required to make this MS suitable for publication in BG.

**Answer**

We sincerely thank the reviewer for taking time to review our manuscript. As the reviewer noticed, the focus of this manuscript is the investigation of changes in phytoplankton community structure over 150 years by using key biomarkers, bulk and compound specific isotopes, quantitative PCR tied to geochemistry, all controlled by hydroclimate variability. Importantly, this study shows the advantages of combining organic geochemistry and molecular studies to constrain natural and anthropogenic influences on lake trophic state over time (i.e. oligotrophic to eutrophic) and the concurrent changes in dominant phytoplankton community dominating the lake. The methodology is likely applicable in many aquatic systems, regardless of whether it is a lake or not. Therefore the focus is more on elucidation the physical and chemical factors that lead to successional changes in dominant microbial community over sub-centennial timescales other than paleolimnology of the lake, whose changes are indeed directly reflected by our data. For example, our data clearly show that the nutrient structure of the lake has changed over the last 150-years because of changes in rainfall and runoff patterns. In turn, this strongly influenced the trophic organization of the lake ecosystem structure. It is not very clear to us what the reviewer has in mind when asking to discuss the lake environment more comprehensively. We have, however, made an attempt to improve on this aspect in the revised manuscript.

Detailed comments: Page 3, line 11: Meaning of the terms "external" and "internal" ecosystem should be specified.

Answer: This is similar to the question posed by the anonymous reviewer #1. We have elaborated more on the external and internal factors in the revised manuscript. By external factors, we mean processes such as rainfall and anthropogenic activities that affect the lake through weathering and runoff intensity. Internal factors include the cycling of various nutrients within the lake, including nutrient regeneration, rates of primary productivity and organic carbon and nitrogen cycling, etc. These are factors, which we subsequently address by the amount of data provided.

Page 5, line 14: Please describe the depth (m) of the sampling location, the "deepest part".

Answer: The deepest part of the lake is approximately 3 m. This has been incorporated in the revised manuscript

Page 5, lines 19-22: I suggest to tone down this part. Robador et al 2015 did not give you off-handed support nor the guarantee for storing sampled sediments without froze them for days. Organic compounds and its isotope compositions can very likely be affected.

Answer: The text has been modified and the reference cited as an example. Samples were kept cool by ice blocks, maintaining maximum temperatures of 4°C. At this temperature the hydrocarbon (the analyzed organic compounds), and let alone their isotope compositions are not likely to be affected significantly. The reviewer may not be aware that such lipid biomarkers and their isotopic compositions are routinely analyzed from million-year-old sediments. Preservation of DNA was more critical, but here we refer to Robador et al (2015), who suggest that heterotrophic microbial activity is severely limited at such temperatures. The fact that our multiple proxies end up showing similar trends for both organic compounds and DNA specific methods substantiates the assumption that any potential degradation has not influenced the main results.

"Such" temperatures needs to be specified.

Answer: The temperature has been specified as 4°C.

Page 5, line 22: It would be better to remove the word "biogeochemical".

Answer: "biogeochemical" has been removed and replaced with "geochemical"

Page 6, lines 3-5: Descriptions of the standard materials for carbon and nitrogen isotope analysis (e.g. working standards) should be included.

Answer: The working standards used for the analysis are:

1) Acetanilide, $\delta^{13}C$=-27.07‰; %C= 71.09%; $\delta^{15}N$=1‰; %N=10.36%.
2) Methionine, $\delta^{13}C$=-26.23‰; %C= 40.25%; $\delta^{15}N$=-2.24‰; %N=9.39%.

These standards were calibrated against standards from IAEA. We have clarified these in the revised manuscript.

Page 6, lines 16-29: It will be better to cite original papers for the method.

Answer: The method has been revised and citations have been added (e.g. Woszcyck et al., 2011; Chawchai et al., 2015, Yamoah, 2016).

Page 7, lines 10-25: For the delta-D analysis, please present at least one set of IRMS chromatogram from analyzed sample.

Answer: A new figure has been added showing an IRMS chromatogram in the revised manuscript.

For the compound specific isotope analysis, especially for the $\delta D$, single-peak baseline separation of targeted compound is essential to get reliable data. It will be better to cite original papers for the analytical method.

Answer: Citations have been added (e.g. Sessions et al., 1999, 2001)

Page 9, line 10-12: Specify the reason to compare sedimentation rates between a lake and a near -by estuary? The two aquatic fields have completely different physical natures, I failed to see the reason or the necessity for their sedimentation rates should be in the same range.

Answer: We agree completely with the reviewer, this comparison has been deleted.

Page11 Sec.4.1: The description of the data trend is rather difficult to understand. Please indicate specific unit name or age from each figures when discussing. Unclear description made it difficult to follow the thread of your discussion. (e.g. p.11 line 22, "the second half of the last century in Figure 3" can be addressed by year).

Answer: The units have been further clarified in terms of ages, as suggested by the reviewer, throughout the manuscript.

Page 11, lines 17-23: This part fails to convince the readers, as some of the discussion seems to be contradictive. I think to draw this conclusion, the $\delta^{15}N$ variation in surrounding watersheds, substrate nitrogen, actual values of phytoplankton and N-fixing cyanobacteria should be considered and discussed. Especially, when the lake is small and easily affected by surrounding environments. The same thing can be said about the discussion regarding $\delta^{13}C$ trends.

Answer: We agree with the reviewer that the lake is easily affected by surrounding environments due to its size. This is however the main reason why we present multiple analysis to really constrain the parameters influencing the dynamics in the lake. The first part of the discussion (Page 11, lines 17-23), which was mainly based on the bulk analysis, show the ambiguity associated with interpretation based solely on bulk parameters especially when inferring biogeochemical cycles back in time.

The importance of this study lies therefore in the combination of proxies and molecular data to elucidate the factors influencing the dominant phytoplankton changes instead of directly measuring the actual values of phytoplankton and N-fixation by cyanobacteria as suggested by the reviewer. In fact, it is impossible to measure such values back 150 year in time. Indeed, one could go to the present-day lake and determine the actual limnological features, however then one still does not know how these were in the past. For deeper timescales, the phytoplankton can only be identified and analysed through proxies recovered from the lake sediments. Moreover, these proxies can be constrained with geochemistry and molecular data. Therefore, we do not think that it is necessary to do direct measurements of the dominant phytoplankton communities that we allude to especially when these factors are properly constrained. We attempt to clarify this aspect better in the revised version.

The reason why the low C/N ratio can be the direct indication to the shift in dominant plankton from diatoms to cyanobacteria should be addressed, too. Answer: The C/N ratio has been used generally in lake systems as a proxy for terrestrial versus aquatic input into lakes. However for a productive lake, changes in phytoplankton community can also change the C/N ratios since different phytoplankton community are modulated by different degree of nutrient enrichment and this would reflect the C/N ratios. Clarity on this will be incorporated in the revised manuscript.

Page 12 Sec.4.2: Recently, Chikaraishi et al. (2012) reported that the terrestrial insects have long-chain n-alkanes ($C_{21}$–$C_{33}$) with lighter $\delta D$ (-195 ± 16‰ abundantly. Does this affect some of your discussion in this section, as the contribution of insect-derived n-alkanes can be one of the reasons for the negative sift in $\delta D$ records?
Answer: Chikaraishi et al., (2012) indeed looked at $\delta D$ of long chain alkanes of terrestrial insects (bees, wasps, and hornets) and had lighter $\delta D$ values (-195 ± 16‰). We however deem it unlikely that n-alkanes from insects would significantly contribute to the total amount of plant-wax derived alkanes, their total biomass is much, much smaller than that of the vegetation. There is not evidence for large amounts of insects deposited in the lake. If insects such as bees, wasps, and hornets dominated the core, one would have expected traces of these insects at least at the topmost part of the core. In addition, $\delta^{13}C$ values from this core (presented in another manuscript) show an entirely terrestrial vegetation origin of the long chain $n$-alkanes ($C_{25}$-$C_{33}$).

Supplement page5 sec 4.3: The English is difficult to understand.

Answer: The text has been rephrased in the revised manuscript

Table S1: Please address units for S, $O_2$, P. Figure S1: Captions should explain all the symbols or lines in the figure. Please remove excess notes.

Answer: Units given in percentages have now been added to the revised manuscript.

[revised manuscript text omitted]

A detailed description of the GC-IRMS program is described in Chawchai et al. (2015).

Instrumental performance and calibration of the reference gas (H$_2$) was achieved by running a standard mixture of $n$-alkanes with a known isotopic composition (reference mixture A4, provided by Arndt Schimmelmann, Indiana University, USA) several times daily. All analyses performed follow that of Sessions et al., (1999, 2001). Results are reported as the weighted mean of triplicate measurements with an average standard deviation of both standards and samples around 4‰ (see Supplement).

**2.5 DNA extraction and qPCR**

[revised manuscript text omitted]

Biogeochemical and biolipid screening of the sediment core, discussed further below, demarcates three distinct periods: *ca.* 2010-1969 CE, *ca.* 1969-1916 CE and *ca.* 1916-1857 CE. The sedimentary deposits are highly organic with TOC contents between 30 and 40%. TOC gradually increases from *ca.* 1857 to 1870 CE and then shows a decrease from *ca.* 1870 to 2008 CE (Fig. 3a). From *ca.* 1857-1970 CE both the $\delta^{13}$C$_{bulk\ org}$ (Fig. 3b) and TN (Fig. 3c) show a gradually increasing trend while $\delta^{15}$N$_{org}$ values rise steadily between *ca.* 1857 and 1969 CE and then decreases from *ca.* 1970-2010 CE (Fig. 3d). The C/N ratio, on the other hand, decreases gradually from the bottom to the top of the sediment core (Fig. 3e). Si/Ti ratio, a marker of silicon input into the lake from terrestrial sources, shows an increase between *ca.* 1916 and 1969 CE (Figs. 3f and g). Scanned photos using ESEM show higher abundances and diversity of diatoms between *ca.* 1916 and 1969 CE (see Supplement; Fig. S6). The P/Ti ratio, which can be used as a proxy for trophic conditions in lakes (Kirilova et al., 2011), show an increase from *ca.* 1970-2010 CE (Fig. 3g).

Biomarkers and qPCR analysis targeting cyanobacteria, diatoms and botryococcus exhibited similar trends, except *ca.* 1970-2010 CE where botryococcenes were not detected although the qPCR analysis detected the presence of *Botryococcus* sp. (Fig. 4) (see supplement for detailed qPCR results). The botryococcene lipid concentrations show an increase between *ca.* 1857 and 1916 CE (Fig. 4a), followed by a decreasing trend from *ca.* 1916-2010 CE. The qPCR data set also shows an increase in *Botryococcus* sp. abundances from *ca.* 1857-1916 CE, a decrease from *ca.* 1916-1969 CE (Fig. 4b), but then again an increase from *ca.* 1970-2010 CE. The HBIs exhibit minimum levels until 1916, after which they show much higher abundance from *ca.* 1916-1969 CE, similar to the diatom abundance as detected by qPCR (Fig. 4c and d), and as seen by ESEM. After 1970 they decrease gradually. The $C_{17}$ *n*-alkanes concentrations show a close resemblance to the *Cyanobacteria* sp. detected by qPCR, with rising concentrations and abundances between *ca.* 1970 and 2010 CE (Figs. 4e and f). The hydrogen isotopic composition of leaf waxes ($\delta D_{wax}$; weighted mean of $\delta D$ $C_{27-31}$ *n*-alkanes) shows a long-term oscillation over the entire 150-year record (Fig. 4g). $\delta D_{wax}$ values have been used as a proxy for rainfall intensity such that lower and higher values indicate an increase and decrease in rainfall intensity, respectively (e.g., Konecky et al., 2013; Niedermeyer et al., 2014). Reconstructed hydroclimatic conditions for southern Thailand over the last 150 years based on $\delta D_{wax}$ values show a relatively drier period from *ca.* 1857-1916 CE and *ca.* 1970-2010 CE and relatively wetter conditions from *ca.* 1916-1969 CE.

**4   Discussion**

The bulk geochemical trends (Fig. 3) indicate that multiple processes control the organic matter (OM) input into Lake NTP. Decreasing C/N ratios with TOC contents of approximately 30% from the bottom to the top of the core suggests organic matter diagenesis, which indicates an aquatic dominated system (Figs. 3a and b). The observed decreasing trend of the C/N ratio, as well as increasing $\delta^{13}C_{org}$ values, may be explained by a preferential (anoxic) mineralization of nitrogen-rich OM, leading to residual OM with a higher C/N ratio (Altabet, 1998; Sachs and Repeta, 1999; Emerson and Hedges, 2003). This explanation would be consistent with the long-term diagenetic OM transformation (Sun et al., 2004). However, successional deposition of different phytoplankton communities, botryococcus (*ca.* 1857-1916), diatoms (*ca.* 1916-1969) and cyanobacteria (*ca.* 1970-2010), with different C/N ratios, $\delta^{13}C_{org}$, and $\delta^{15}N_{org}$ can also influence the bulk profiles.

An alternate interpretation of the increasing $\delta^{13}C_{org}$ values from the bottom to the top of the core lies in a successional shift of the trophic status of the lake (Brenner et al., 2000; Meyers and Teranes, 2001). Increasing nutrient levels promote primary productivity, thereby depleting the amount of dissolved inorganic carbon. This could lead to lower net fractionation against $^{13}C$ by the phytoplankton during photosynthesis, thereby increasing the $\delta^{13}C_{org}$ values (Meyers and Teranes, 2001).

Observed variations of dominant phytoplankton community over time can be explained best by changes in the trophic level of the lake. The period from *ca*. 1857 to 1916 CE is marked by significant increases in both *Botryococcus* sp. abundance (Fig. 4a) and botryococcene lipids (Fig. 4b), which also corresponds with slightly lower precipitation (Fig. 4g). *Botryococcus*

*braunii* has been suggested to thrive in relatively oligotrophic conditions (Souza et al., 2008)

and can, therefore, be used as a proxy for such conditions within the oxygenated epilimnion (Waldmann et al., 2014). Although the highly organic sediment may suggest a productive lake, at variance with oligotrophic lake water conditions, we argue that the final TOC content in lake sediments, especially in the tropics, is more determined by OM preservation than by the incoming flux. In the tropics, the mean air temperature (MAT) is a direct result of incoming solar radiation, and the relationship between the MAT and the amount of rainfall are typically inversely proportional (Imboden and Wüest, 1995; Boehrer and Schultze, 2008).

Dry, cloudless and warmer conditions lead to stronger stratification in freshwater lakes (Imboden and Wüest, 1995). We hypothesize that a long-term decrease in rainfall and slightly higher temperatures would result in a thermal stratification of the lake. Under stratified conditions reformed nutrients would remain in the anoxic hypolimnion. As a result, the surface water remained relatively oligotrophic enabling *Botryococcus braunii* to thrive whereas the bottom of the lake was characterized by bacterial breakdown of organic N. This argument is supported by increasing $\delta^{15}N_{org}$ values from *ca*. 1857 to 1916 CE, which suggests extensive denitrification in the lake sediment, as denitrification typically leads to more depleted $\delta^{15}N$ values in the residual organic matter (Wada, 1980).

A stark difference in the dominant phytoplankton community is observed from *ca*. 1916-1969

CE. This period is marked by significant decrease in *Botryococcus* sp. gene abundance (Fig.

4a) and botryococcene lipid concentrations (Fig. 4b) while diatom abundance (Fig. 4c) and

$C_{25}$ HBIs concentrations (Fig. 4d), which is a useful indicator of diatom-derived OM inputs to sediments (McKirdya et al., 2013) increases. More directly, we observed a marked increase in diatom diversity on the ESEM image scans of the sediments (Supplement; Fig. S6). Diatoms dominate phytoplankton communities as long as there is abundant silica irrespective of changes in environmental conditions and nutrient levels (Egge and Aksnes, 1992). Interestingly, the increase in diatom markers (Fig. 4c and d) coincides with an increase in reconstructed rainfall intensity (Fig. 4g). Moreover, the increase in Si/Ti ratios (Fig. 3f), a run-off signal (Murphy et al., 2000), coincides with high diatom blooms, in particular between $ca.$ 1916 and 1969 CE. Since Ti is a highly immobile element, weathering and transportation of Si is not accompanied by significant Ti delivery to aquatic basins. Therefore, the Si/Ti ratio can serve as a proxy for nutrient dynamics linked to hydrological changes (e.g. Cartapanis et al., 2014) and as an indicator for enhanced diatom production in lakes (Wennrich et al., 2014). Altogether, it appears that the wetter conditions between $ca.$ 1916 and 1969 CE increased catchment runoff into the lake, leading to elevated nutrient and silicate mineral flux to the lake water (e.g. Paerl et al., 2006). In line with the argument above, it is also well possible that an increase in precipitation was accompanied by cooler temperatures, which then led to a decrease in thermal stratification and an increase in mixing between the epilimnion and hypolimnion, making reformed nutrients from the bottom waters available at the lake water surface.

After $ca.$ 1969 CE, the phytoplankton community structure changed again, with a diminishing role for diatoms as evidenced by lower concentrations of $C_{25}$ HBIs (Fig. 4c) and the start of a marked increase in cyanobacteria gene numbers (Fig. 4e) and $C_{17}$ $n$-alkane concentrations (Fig. 4f). $C_{17}$ $n$-alkanes are recognized biomarkers of aquatic algae and photosynthetic bacteria such as cyanobacteria (Meyers, 2003). Indeed, cyanobacteria gene numbers relative to bacteria quantification based on the qPCR data covary strongly with $C_{17}$ $n$-alkane concentration, which indicates that the $C_{17}$ $n$-alkanes were likely produced by cyanobacteria. Additionally, the decreasing $\delta^{15}N_{bulk\ org}$ values during this period (Fig. 3d) also suggest nitrogen fixation, a process strongly associated with cyanobacteria blooms (Vahtera et al., 2007). The most recent period ($ca.$ 1969-2010 CE) is also characterized by somewhat lower rainfall amounts (Fig. 4g). The amount of Si indeed decreased, and diatoms became less abundant, allowing non-diatomaceous phytoplankton not dependent on Si to take over (Egge and Aksnes, 1992).

The number of botryococcus genes increased again (Fig. 4a). However, this was not accompanied by the production of botryococcene lipids. This mismatch can be the result of the primer used for the quantification of *Botryococcus braunii* not being specific enough. *Botryococcus braunii* are classified into three races: A, B, and L, and it is only the B race that produces botryococcenes (Eroglu et al., 2011) whereas the A and L races produce long-chain alkadienes and lycopadienes, respectively (Metzger and Largeau, 2005). A change in race thus may explain the disappearance of botryococcenes from the upper part of the sediment record. Since the qPCR primers for botryococcus were not specific enough, it is also possible that it picked up closely related green algae that took over the ecological niche of *B. Braunii* under different trophic conditions. The absence of long-chain alkadienes and lycopadienes in the upper sediment layers supports this argument and indicate that conditions had indeed become unfavorable to *B. Braunii*, in a similar way as described by Smittenberg et al. (2005).

Phosphorus bioavailability is one of the most important factors limiting aquatic cyanobacteria blooms (Paerl and Fulton, 2006; Paerl and Valerie, 2012), and the shift to cyanobacterial prevalence thus suggests eutrophic phosphorous-rich conditions, instead of the oligotrophic-like conditions that occurred a century earlier under otherwise similar climatic conditions. Indeed, higher levels of phosphorus were detected in the upper part of the sediment (Fig. 3g). The source of the elevated phosphorus is unclear but has likely resulted from human activities. Under a land development program in the 1990s, more than 20% of Thailand's 56,000 villages were located in forest reserves (Gray, 1991; Puri, 2006), which allowed the expansion of land encroachment and agricultural activities. For instance, southern Thailand has seen an increase in the cultivation of rubber trees on small farms at rates above 7% $yr^{-1}$ (Leturque and Swiggings, 2011). The use of fertilizers during farming activities, untreated wastewater effluents and the use of detergents are likely sources of the elevated phosphorus inputs into the lake (Litke, 1999; Chislock et al., 2013). These accelerated the eutrophic state of the lake beyond the natural rate of nutrient enrichment, which takes centuries to achieve (Litke, 1999).

According to our analysis, photosynthetic primary production is at the base of internal organic matter production in the lake. However, changes in precipitation and anthropogenic nutrient input have produced fluctuations in dominant primary producer communities over the last 150 years, from botryococcene lipid-producing algae to diatoms to a dominance of cyanobacteria. Overall, we suggest that NTP is sensitive to environmental stressors and that multiple processes discussed above control the OM input and carbon storage. The bulk geochemical analysis was clearly inadequate to disentangle the processes that have been controlling the limnological, ecological and microbial dynamics in NTP. Instead, the combination of bulk geochemistry, lipid biomarkers, and molecular DNA analysis could appropriately constrain many of these processes.

**5   Summary and conclusion**

Validation of paleoenvironmental and paleoclimatic proxies is necessary to constrain geochemical patterns through time. The coupled lipid biomarker and qPCR data allowed different microbial groups in the lake sediments to be distinguished and quantified, leading to the identification of meaningful biological relationships between the phytoplankton community structure response to either anthropogenic or natural environmental changes over the last 150 years.

Our results show that between *ca.* 1857 and 1916 CE, relatively drier climate in southern Thailand coincided with relatively oligotrophic surface water conditions in Lake NTP, which allowed *Botryococcus braunii* species to bloom. From *ca.* 1916-1969 CE, an increase in precipitation resulted in higher Si input into the lake, which led to a rapid takeover by diatoms as primary producers. Since the 1970s, many aspects of the initial limnic state returned upon drier conditions, except that anthropogenic impact led to an increase in phosphorus thereby allowing cyanobacteria to become a major contributor to primary productivity. Although the qPCR method did again detect genetic evidence for the presence of *Botryococcus* sp. over the last decades, the specific lipid biomarkers for *Botryococcus* sp. were not found anymore, which can be due to limitations of either proxy.

We conclude that the combination of geochemical and lipid biomarker proxies together with qPCR analyses is a useful approach that has the potential to assist in tracking the effects of changing climate on primary producers and also to assess these effects on the carbon cycle. This multi-proxy approach may help refine the knowledge about the use and shortcomings of the different proxies, which is critical for their interpretation especially when used on a more stand-alone basis.

[revised manuscript text omitted]

sediment) (blue dashed), (b) botryococcenes, a proxy for *Botryococcus braunii* (B race) (blue solid), (c) diatoms (%) (purple dashed), (d) $C_{25}$ Highly branched Isoprenoid (HBIs), a proxy for diatoms (red solid), (e) cyanobacteria (%) (mustard dashed), (f) $C_{17}$ *n*-alkane, a proxy for cyanobacteria (brown solid) and (g) δD values of $C_{27-29-31}$ *n*-alkanes (δD$_{wax}$), a proxy for rainfall amount (green). The black dashed lines represent the transitions between the different units. For interpretation of the references to color, the reader is referred to the web version of this article.